# Lifesaving Treatment for DISH Syndrome in the Tenth Decade of Patient’s Life

**DOI:** 10.3390/geriatrics10040092

**Published:** 2025-07-07

**Authors:** Bartosz Krolicki, Victor Mandat, Tomasz S. Mandat

**Affiliations:** 1Department of Neurosurgery, Maria Sklodowska-Curie Memorial Oncology Institute, 02-781 Warsaw, Poland; 2Department of Biology, University of Toronto Mississauga, Toronto, ON L5L 1C6, Canada

**Keywords:** Forestier–Rotes-Querol syndrome, DISH, dysphagia, dyspnea, elderly

## Abstract

**Background/Objectives**: Diffuse idiopathic skeleton hyperostosis (DISH) is also known as Forestier–Rotes-Querol syndrome. The etiology of DISH is unknown. DISH is characterized by ossification of the anterior longitudinal ligaments of the spine. The area most frequently involved in the disease is the thoracic region of the spine. DISH in most cases is asymptomatic. If the cervical spine is involved, the most common symptoms are dysphagia and dyspnea. The ossifications in the cervical region of the spine are localized most frequently in its lower segments. **Case presentation**: The authors present the case of a 92-year-old cachectic female patient (body mass index (BMI) of 17; lost 13% of her body weight within the last 6 months). The patient underwent resection of the anterior osteophytes C2-T1. Results: At one-year follow up, the patient had gained weight (BMI—20) and regained her ability to consume solid products. To our knowledge, this is the oldest patient treated surgically for DISH. **Conclusions**: If dysphagia or dyspnea appears among elderly patients, cervical spine inspection should be conducted. If DISH is diagnosed safe, effective surgical treatment should be considered.

## 1. Introduction

Diffuse idiopathic skeleton hyperostosis (DISH) is also known as Forestier–Rotes-Querol syndrome [1]. DISH is a condition characterized by ossification of the anterior longitudinal ligaments along the spine, often involving multiple levels. Frequently, it presents with spinal stiffness and intermittent pain. In contrast to other spinal conditions like spondyloarthropathies, in DISH there is an absence of inflammatory arthritis. The etiology of DISH is unknown. The area most frequently involved in the disease is the thoracic region of the spine. The ossifications in the cervical region of the spine are localized most frequently in its lower segments. DISH in most cases is asymptomatic. If the cervical spine is involved, the most common symptoms are dysphagia and dyspnea [2,3]. The causes of dysphagia and dyspnea are rarely sought in spinal deformity and patients are often left untreated. Severe dysphagia and dyspnea can significantly affect quality of life and might shorten the lifespan of patients. Drawing attention to problems that might be safely, successfully, and minimally invasively solved among the majority of patients regardless of their age might positively affect the quality of life of a significant number of elderly patients.

## 2. Case Description

We present the case of a 92-year-old generally healthy, socially independent female patient diagnosed with DISH (according to Resnick’s criteria) [4]. The patient complained of difficulty in swallowing solid products for several years. The patient was treated because of dysphagia and choking for two years prior to admission. Gastroscopy did not reveal any abnormalities. MRI showed DISH at the C3-T1 level. One month before admission, the symptoms increased and the patient could only consume liquid products. Her body mass index (BMI) decreased to 17 (weight: 43 kg; height: 159 cm; lost 13% of her body weight within the last 6 months). Extensive ossification of the anterior longitudinal ligaments starting from C2 and extending through the whole cervical spine up to T1 vertebrae encroaching the esophagus was detected upon a CT scan of the spine (Figure 1). Due to increasing symptoms of dysphagia, the patient underwent surgery after signing the consent form for the surgery. Using a standardized approach for anterior cervical disk fusion, the C2-T1 osteophytes were resected with a high-speed drill. Because of the patient’s advanced age and the fact that the cervical spine was stable, the disks were not removed, nor was fusion or instrumentation added to the procedure. There were no postoperative complications. The surgery lasted 42 min. On the day following surgery, the patient was mobilized and could eat soft food. No cervical immobilization was used following surgery. On the second day after surgery, the patient was discharged home. Phisiotherapy was started on the 14th day after surgery. At 30-day follow-up, the patient reported that her swallowing had improved, and the patient could consume all forms of food. Post-operative CT (Figure 2) and MRI (Figure 3) scans were performed, revealing that osteophytes had been extensively removed and that the esophagus was decompressed. At one-year follow up, the patient had gained weight (BMI—20) and regained her ability to consume solid products. The quality of patient’s life measured with EQ-5D-5L improved by 61% (18 to 7) (EuroQol Research Foundation).

## 3. Discussion

Even though the etiology of DISH is unknown, recent reports indicate an association between DISH and metabolic disorders. The most common are diabetes, hyperinsulinemia, obesity, and dyslipidemia [5]. The presented patient did not suffer from any chronic or metabolic disease.

In the elderly population, anterior cervical osteophytes are commonly identified. During autopsies, 28% of the analyzed elderly subjects were found to have morphological symptoms of DISH. Forrestier syndrome occurs in 25% of the male population aged over 50 years and 15% of females in the same age group. Recent reports indicate that the first morphological changes identified on CT or MRI scans can be identified in the fourth decade of life [2]. Those changes are usually clinically asymptomatic. The main indication or contraindication for surgical treatment of a degenerative spine is the biological age, making the metrical age less significant [5,6]. Most spine centers do not consider patients for corrective surgery in the ninth decade of life. The presented patient suffered from dysphagia for several years from the ninth to the tenth decade of life. Because of her life-threatening dysphagia that resulted in cachexia, this patient in her tenth decade of life qualified for a life-saving surgical procedure.

Most commonly, the C5–C6 vertebrae are involved in DISH (40%). Osteophytes are less frequently observed at the C4–C5 level, and only 14% are observed at the C2–C3 level [7,8]. When the upper levels are involved, the respiratory symptoms tend to dominate because the osteophytes impinge the hypopharynx. Involvement of the lower cervical region usually results in impingement on the esophagus and dysphagia [5,6]. Patients with multilevel osteophytes can present with both respiratory and digestive tract symptoms. Dysphagia is the most common symptom, affecting 28% of patients. The presented patient reported severe dysphagia. Even though the upper cervical spine was involved, the patient did not report any respiratory problems. The patient underwent selective and minimally invasive surgery when the osteophytes were removed. Von der Hoeh et al. recommend ACDF for patients with DISH and concomitant neck pain, instability, or spinal stenosis. The authors believe that the lack of those additional symptoms prevent the expansion of the scope, especially among elderly patients [9,10].The problem related to reossification might be substantial for younger patients and was reported at two years follow-up. The findings observed on X-rays did not lead to recurrence of the symptoms in the analyzed group [10].

## 4. Conclusions

Usually, elderly patients who report breathing problems or dysphagia are not referred to spine centers for surgical treatment. If the endoscopic evaluation does not reveal any mechanical problems, the changing of meal habits or swallowing training is usually recommended, and in most cases, patients are not referred for neuroimaging studies. If DISH syndrome is diagnosed, the surgical removal of osteophytes should be recommended to all age groups, which might significantly improve the patient’s quality of life.

## Figures and Tables

**Figure 1 geriatrics-10-00092-f001:**
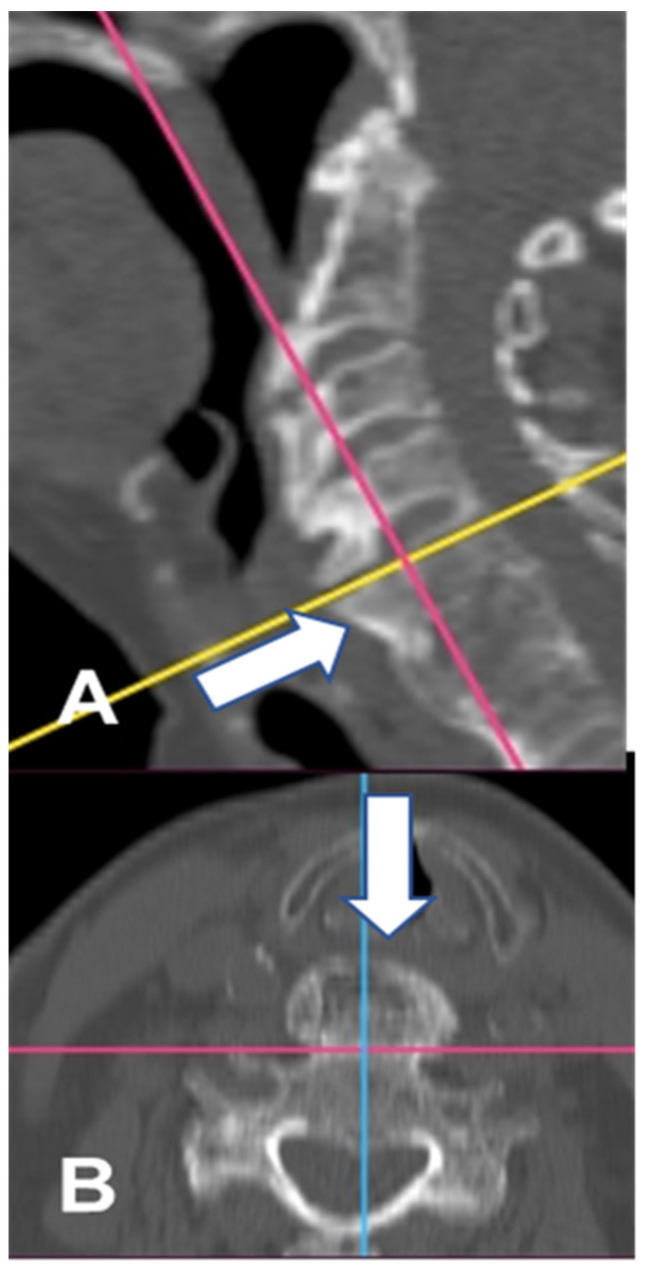
CT images (bone window) of the cervical spine prior to surgery. Osteophytes are visualized on the sagittal (**A**) and axial (**B**) planes (white arrow). The yellow line indicates the plane parallel to the upper C5 vertebral lamina. The pink lines indicate a plane parallel to the anterior part of the vertebrae body potentially without osteophytes. The blue line indicates the midline of the C5 vertebral body.

**Figure 2 geriatrics-10-00092-f002:**
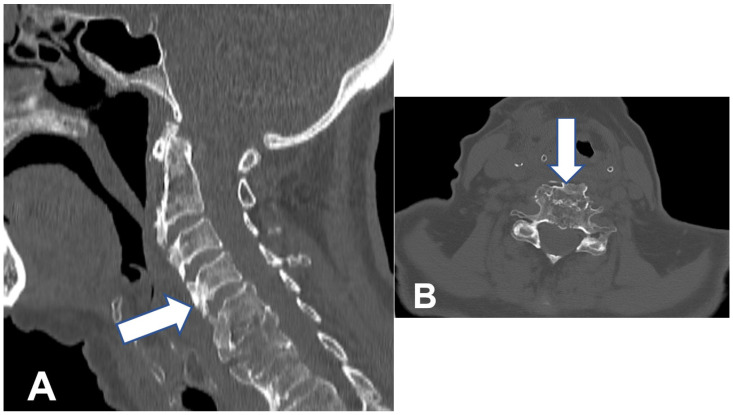
CT images(bone window) of the cervical spine after surgery on the sagittal (**A**) and axial (**B**) planes. Liberated paravertebral structures are visualized (white arrow).

**Figure 3 geriatrics-10-00092-f003:**
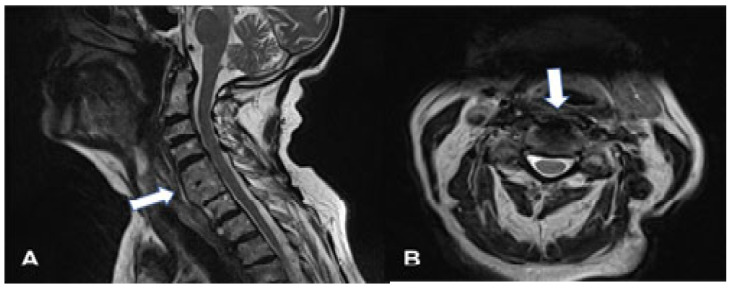
MRI images (T2) of the cervical spine after surgery on the sagittal (**A**) and axial (**B**) planes. Liberated paravertebral structures that include the esophagus are visualized (white arrow).

## Data Availability

The original contributions presented in this study are included in the article. Further inquiries can be directed to the corresponding author.

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
