# Peer review of "Lifesaving Treatment for DISH Syndrome in the Tenth Decade of Patient’s Life"

_geriatrics, 2025, doi:10.3390/geriatrics10040092_

Round 1

Reviewer 1 Report

Comments and Suggestions for Authors

The Lifesaving Treatment for DISH Syndrome in the Tenth Decade of a Patient’s Life reports a remarkable case of a patient in her 90s diagnosed with DISH syndrome. The patients exhibited extensive ossification along the cervical spine, which compressed the esophagus. The ossified mass was removed using a drill. The procedure was tolerated well despite the patient’s advanced age. And the patient’s outcome was good with a significant improvement in symptoms postoperatively.

1, Figures 1 and 2 form a paired set. Please present the preoperative and postoperative CT images side by side. Use arrows to label the DISH position before and after surgery. Therefore, the morphological changes before and after surgery can be clearly demonstrated.

2, Please label the esophagus and illustrate how it was decompressed following surgery, highlighting the relief of compression compared to the preoperative state.

  1. Use the same idea to label Figure 3.
Comments on the Quality of English Language

Fine.

Author Response

1, Figures 1 and 2 form a paired set. Please present the preoperative and postoperative CT images side by side. Use arrows to label the DISH position before and after surgery. Therefore, the morphological changes before and after surgery can be clearly demonstrated.

Response: The figures 1,2 and 3 were corrected.

  1. Please label the esophagus and illustrate how it was decompressed following surgery, highlighting the relief of compression compared to the preoperative state.

Response: corrected

  1. Use the same idea to label Figure 3.

Response: corrected

The esophagus is visible on the MRI, postoperative images, but clear distinction is not possible. On the CT scans trachea is visible. The labeling arrows were added.

Reviewer 2 Report

Comments and Suggestions for Authors
  1. Please recheck abbreviation management in the entire document: abbreviate a term the first time you use it, then use only the abbreviation.
  2. “MRI showed DISH at the C3-Th1 level” – what does “Th1” mean? Do you mean “thoracic”? Please just leave “T1” since “Th1” can be confusing (for example with T helper 1 cells).
  3. “Due to increasing symptoms of dysphagia patient underwent surgery.” – please confirm in the manuscript that the patient gave written informed consent for surgery and publication.
  4. “Due to increasing symptoms of dysphagia patient underwent surgery.” – You only specify the region and the drill. Please offer more surgical details: surgical approach, procedure details (intraoperative findings, duration and anesthesia details, deviations or complications during surgery), postoperative care (analgesia, immobilization, rehabilitation).
  5. Figure 1 – please use a better image quality in terms of resolution.
  6. Figure 1, Figure 2 and Figure 3 actually contain 2 images each, please mark them with “a” and “b” and explain in the figure caption what each of them is (for example incidence, anatomical region).
  7. Figure 1, Figure 2 and Figure 3 – please mark on the image using some sort of symbol (for example *) the pathological elements or their absence.
  8. All figures – please include proper imaging reporting information: specify plane (sagittal, axial, coronal), for MRI indicate time (T1, T2, STIR, DWI, contrast-enhanced), for CT indicate window (bone window, soft tissue window, or contrast-enhanced).
  9. If feasible, include the patient’s viewpoint on diagnosis and treatment.
  10. If allowed, please include an intraoperative photo.
  11. Please include in the Discussion at least another article that reports surgery for DISH and cite it.

Author Response

  1. Please recheck abbreviation management in the entire document: abbreviate a term the first time you use it, then use only the abbreviation.

Response: The abbreviations were corrected (Case description)- like abbreviations like MRI or CT were not extended.

  1. “MRI showed DISH at the C3-Th1 level” – what does “Th1” mean? Do you mean “thoracic”? Please just leave “T1” since “Th1” can be confusing (for example with T helper 1 cells).

Response: Th1 were changed to T1

  1. “Due to increasing symptoms of dysphagia patient underwent surgery.” – please confirm in the manuscript that the patient gave written informed consent for surgery and publication.

Response: The information about consent was added to the manuscript

The information about consent was added to the manuscript (case description section).

  1. “Due to increasing symptoms of dysphagia patient underwent surgery.” – You only specify the region and the drill. Please offer more surgical details: surgical approach, procedure details (intraoperative findings, duration and anesthesia details, deviations or complications during surgery), postoperative care (analgesia, immobilization, rehabilitation).

Response: The information regarding surgery, complication, mobilisation and rehabilitation was added to the manuscript.

  1. Figure 1 – please use a better image quality in terms of resolution.

Response:  The figures were corrected according to Reviewer #1 and 2 indications.

  1. Figure 1, Figure 2 and Figure 3 actually contain 2 images each, please mark them with “a” and “b” and explain in the figure caption what each of them is (for example incidence, anatomical region).

Response: The figures were corrected according to Reviewer #1 and 2 indications.

  1. Figure 1, Figure 2 and Figure 3 – please mark on the image using some sort of symbol (for example *) the pathological elements or their absence.

Response: The figures were corrected according to Reviewer #1 and 2 indications.

  1. All figures – please include proper imaging reporting information: specify plane (sagittal, axial, coronal), for MRI indicate time (T1, T2, STIR, DWI, contrast-enhanced), for CT indicate window (bone window, soft tissue window, or contrast-enhanced).

Response: The figures were corrected according to Reviewer #1 and 2 indications.

  1. If feasible, include the patient’s viewpoint on diagnosis and treatment.

Response: The information regarding qol was added to the Case description section.

  1. If allowed, please include an intraoperative photo.

Response:  The surgery was not recorded- we are unable to provide a photograph.

  1. Please include in the Discussion at least another article that reports surgery for DISH and cite it.

Response:  von der Hoeh NH, Voelker A, Jarvers JS, Gulow J, Heyde CE. Results after the surgical treatment of anterior cervical hyperostosis causing dysphagia. Eur Spine 2015;24(Suppl 4):S489–93. and Scholz C, Naseri Y, Hohenhaus M, Hubbe U, Klingle J-H. Long-term results after surgical treatment of diffuse idiopathic skeletal hyperostosis (DISH) causing dysphagia. Journal of Clinical Neuroscience 67 (2019) 151–155. Were added to the references.

Reviewer 3 Report

Comments and Suggestions for Authors

Diffuse idiopathic skeletal hyperostosis (DISH) is a type of arthritis that leads to the stiffening and calcification of soft tissues, including ligaments, tendons, and joints. Furthermore, the patient may also develop bone spurs, which have been observed to cause pain and other symptoms. Consequently, the subject matter of the present article is pertinent and may pique the interest of readers of the journal Geriatrics, as this malady is predominantly observed in elderly patients. Nonetheless, I would like to offer the following observations on the design and content of this article.

(1) Abstract: In the abstract, it is not customary to give the numbers of literary references. A more systematic approach to the description of the abstract is warranted, and the following clinical case can serve as a model: Galloway, M.; Hoffman, N.; Bray, C.L.; Ebrahim, A.; Puebla, B.; Ritchie, D. Case Report: Weakness and Recurrent Falls in an Older Patient. Geriatrics 2025, 10, 41. https://doi.org/10.3390/geriatrics10020041

(2) In the above example, the authors can familiarize themselves with the customary practice of issuing references in accordance with the MDPI style.

(3) Introduction: DISH is not an uncommon pathology in the elderly. In such instances, it is recommended that the authors provide an explanation for the clinical case's distinctiveness and the value of its description for clinical practice.

(4) Case Description: It is recommended that authors present not only the underlying disease of a 92-year-old patient but also a brief analysis of concomitant diseases, with characteristics of this patient.

(5) At the conclusion of the article, it is recommended that the authors incorporate the "Authors Contributions" section.

Author Response

Diffuse idiopathic skeletal hyperostosis (DISH) is a type of arthritis that leads to the stiffening and calcification of soft tissues, including ligaments, tendons, and joints. Furthermore, the patient may also develop bone spurs, which have been observed to cause pain and other symptoms. Consequently, the subject matter of the present article is pertinent and may pique the interest of readers of the journal Geriatrics, as this malady is predominantly observed in elderly patients. Nonetheless, I would like to offer the following observations on the design and content of this article.

(1) Abstract: In the abstract, it is not customary to give the numbers of literary references. A more systematic approach to the description of the abstract is warranted, and the following clinical case can serve as a model: Galloway, M.; Hoffman, N.; Bray, C.L.; Ebrahim, A.; Puebla, B.; Ritchie, D. Case Report: Weakness and Recurrent Falls in an Older Patient. Geriatrics 2025, 10, 41. https://doi.org/10.3390/geriatrics10020041

Response: the Abstract was corrected

(2) In the above example, the authors can familiarize themselves with the customary practice of issuing references in accordance with the MDPI style.

Response: 2,3,4 The information was added to the Case description section.

(3) Introduction: DISH is not an uncommon pathology in the elderly. In such instances, it is recommended that the authors provide an explanation for the clinical case's distinctiveness and the value of its description for clinical practice.

Response: The information was added to the introduction section.

(4) Case Description: It is recommended that authors present not only the underlying disease of a 92-year-old patient but also a brief analysis of concomitant diseases, with characteristics of this patient.

Response: The information was added to the Case description section.

(5) At the conclusion of the article, it is recommended that the authors incorporate the "Authors Contributions" section.

Response: the section was added to the manuscript body.